# Retrieving Performances of Vortex Beams with GS Algorithm after Transmitting in Different Types of Turbulences

**Maxime Irene Dedo**[ID]**, Zikun Wang, Kai Guo**[ID]**, Yongxuan Sun, Fei Shen, Hongping Zhou, Jun Gao, Rui Sun**[ID]**, Zhizhong Ding and Zhongyi Guo** *[ID]

School of Computer and Information, Hefei University of Technology, Hefei 230009, China; dedomax@163.com (M.I.D.); wangzikun421@163.com (Z.W.); kai.guo@hfut.edu.cn (K.G.); syx@hfut.edu.cn (Y.S.); shenfei@hfut.edu.cn (F.S.); ciangela@hfut.edu.cn (H.Z.); gaojun@hfut.edu.cn (J.G.); sunrui@hfut.edu.cn (R.S.); zzding@hfut.edu.cn (Z.D.)
* Correspondence: guozhongyi@hfut.edu.cn; Tel.: +86-186-5515-1981

**Abstract:** The transmission of the orbital angular momentum (OAM) beam has attracted a lot of attention in the field of free-space optical (FSO) communication. Usually, after transmitting in atmospheric turbulences, the helical phase-front of OAM beams will be severely distorted, and there will exist the intermode crosstalk. As a result, the performance of the communication system will degrade significantly. In this paper, we have investigated the influences of the level of the turbulence strength to the transmitting OAM beams by changing the refractive-index structural parameter of $C_n^2$ and the number of turbulence random phase screens of $N$ in simulation environment. Then, by adopting the Gerchberg-Saxton (GS) algorithm, which can be used to compute the pre-compensation phase and correct the distorted OAM beams, the retrieving performances of transmitting single and multiplexed OAM beams under different turbulence strengths were also investigated. The simulation results show that with increasing the atmospheric-turbulence strength levels determined by the parameters $C_n^2$ and $N$, the retrieving performances decrease dramatically. When the turbulence strength level is selected within an appropriate range, the OAM beams can be effectively retrieved by adopting GS algorithm and observing the power density spectrum. Notably, the retrieving performance for the transmission of a single OAM beam is better than that of the multiplexing OAM beam.

**Keywords:** orbital angular momentum; free-space optical communication; atmospheric turbulence; atmospheric correction

## 1. Introduction

In 1992, Allen et al. demonstrated that the Laguerre-Gaussian (LG) beam with helical wave-front carries the orbital angular momentum (OAM) [1]. It was shown that the LG beam with helical phase term of $\exp(il\phi)$, have an OAM of $l\hbar$ per photon, where $\phi$ is the azimuthal angle, is the Planck's constant divided by $2\pi$, and $l$ is the azimuthal index called topological charge which represents the OAM mode. A key property of the OAM beams is that beams with different topological charges are mutually orthogonal. As a result, the beams can be effectively multiplexed and de-multiplexed to increase the capacity of high-speed optical communication, including free-space optical (FSO) and optical fiber communication [2–11]. In addition, OAM has been widely applied to many other fields, such as optical tweezers [12,13], quantum information processing [14–16], super high-density data storage [17], and so on.

For a practical OAM-based FSO communication link, atmospheric turbulence is a serious challenge, which distorts both intensity and phase of the received light field, because it will introduce intermode

crosstalk, thereby degrading the performance of the communication system [18–24]. Thus, compensation of the distorted optical field becomes necessary and important. Several compensation techniques, such as MIMO equalization [25], adaptive optics system [26–29] and phase compensation based on Gerchberg-Saxton (GS) algorithm [30,31], have been preliminarily explored in the past years. However, due to the random effect of the atmospheric turbulence, the variation trend of turbulence, which is associated with the refractive index, structure parameter and the number and location of simulated random phase screen insertion in the transmission distance, are still worthy of detailed discussion.

In this paper, we have emulated the influencing effects of the atmospheric turbulence on the transmitting single and multiplexed OAM beams by inserting random phase screens into the transmission path with different turbulence strength levels by changing the refractive-index structural parameter of $C_n^2$ and the number of phase screens of $N$. Then, the GS algorithm is adopted to calculate the pre-compensation phase and correct both intensity and phase of transmitted LG beams, from which the OAM beams can be effectively retrieved.

## 2. Materials and Methods

### 2.1. The Description of the Orbital Angular Momentum

The LG beam, one of optical vortex (OV) beams carrying the OAM, is widely used in OV-based communication system because it is relatively easy to produce in experimental environments. The original transmitted LG beam in the electrical field can be expressed in cylindrical coordinates as [32,33]

$$
\begin{aligned}
LG_{lp} &= \sqrt{\frac{2p!}{\pi(p+|l|)!}} \frac{1}{\omega(z)} \left[\frac{r\sqrt{2}}{\omega(z)}\right]^{|l|} \exp\left[\frac{-r^2}{\omega^2(z)}\right] L_p^{|l|}\left(\frac{2r^2}{\omega^2(z)}\right) \\
&\times \exp\left[\frac{-ikr^2 z}{2(z^2+z_R^2)}\right] \exp\left[i(2p+|l|+1)\tan^{-1}\left(\frac{z}{z_R}\right)\right] \exp(il\phi),
\end{aligned}
\tag{1}
$$

where $r$ is the distance from the propagation axis, $\varphi$ is the azimuthal angle, $z$ is the distance along the propagation axis, $k = 2\pi/\lambda$ is the wave number, $\lambda$ is the wavelength, $l$ denotes the topological charge which is an integer representing OAM mode, $p$ is radial index (in our simulation we consider $p = 0$), the term $L_p^l(.)$ is the Laguerre polynomial, $\omega(z) = \omega_0 \sqrt{1+(z/z_R)^2}$ is the Gaussian beam radius at distance $z$ where $\omega_0$ is the beam waist, $z_R = \pi\omega_0^2/\lambda$ is the Rayleigh range, and the phase profile expressed as $\exp(il\phi)$ is what allows these beams to exhibit OAM.

### 2.2. The Model of Atmospheric Turbulence

In this paper, a Kolmogorov turbulence phase screen model [20,24] is taken into account to simulate the atmospheric turbulence. Its power spectrum function can be expressed as [34,35]:

$$
\Phi(K) = 0.033 C_n^2 K^{-11/3}
\tag{2}
$$

where $L_0$ and $l_0$ represent the outer turbulence radius and the inner turbulence radius respectively. $C_n^2$, as the atmospheric refractive-index structure parameter, characterizes the intensity of atmospheric turbulence. It describes the variation of the refractive index between two points. The refractive index can also be determined from temperature and pressure. In addition, three important parameters, such as the Rytov variance $\sigma_R^2$, the overall strength of turbulence $D/r_0$ and the Fresnel number $N_f = D/(\lambda L)$ are selected to emulate the atmospheric turbulence [22], where $L$ represents the link distance, $D$ denotes the aperture sizes of the link and $\lambda$ is the wavelength. Atmospheric refractive index is related to the Fried coherence length $r_0$ which can be determined using following equation [35]:

$$
r_0 = \left[\frac{2.91}{6.88}k^2 \int\limits_0^L C_n^2(z)dz\right]^{-3/5}.
\tag{3}
$$

Assuming that $C_n^2(z)$ is a constant value along the transmission path, $C_n^2(z) \equiv C_n^2$, $r_0$ can be simplified as $r_0 = 0.185\left[\lambda^2/\left(C_n^2 Z\right)\right]^{3/5}$, where $Z$ is the transmission distance. In numerical simulation, the long channel under atmospheric turbulence can be divided into $N$ successive sub-channels and can be simulated by inserting random phase screen in the middle of each sub-channel. Then $r_0$ can be written as $r_0 = 0.185\left[\lambda^2/\left(C_n^2 \times \Delta z\right)\right]^{3/5}$. To measure the scintillation, the Rytov variance $\sigma_R^2$ is given by the following expression

$$\sigma_R^2 = 1.23 k^{7/6} C_n^2 L^{11/6} \tag{4}$$

In this way, a kolmogorov turbulence phase screen can be expressed as

$$D_\phi(r) = 6.88(r/r_0)^{5/3} \tag{5}$$

where $\phi$ is assumed to be the distortion phase caused by the turbulence after the light beam transmits through the turbulence phase screen. Figure 1 shows the schematic demonstration of the atmospheric turbulence channel with $N$ random phase screens.

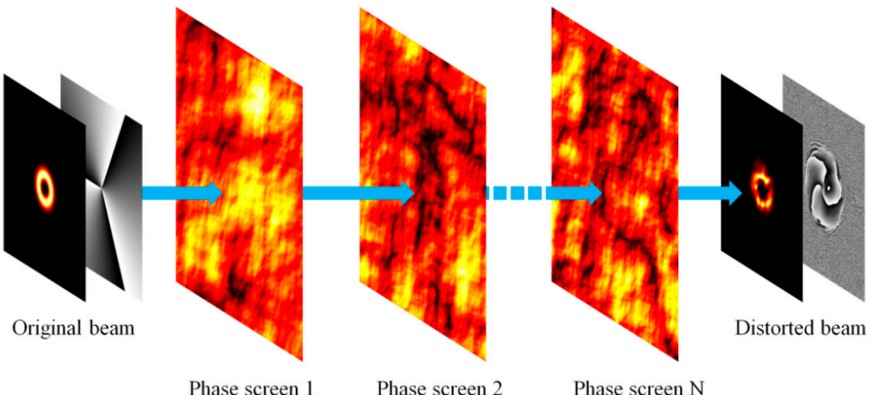

**Figure 1.** Schematic demonstration of atmospheric turbulence channel with $N$ random phase screens. The distance $\Delta z$ between each screen is $Z/N$. ($Z$: transmission distance; $N$: the number of sub-channels).

### 2.3. The Theory of GS Algorithm

The Gerchberg-Saxton (GS) algorithm is well known as an iterative algorithm that has received a lot of attention in the field of phase information retrieval [36], beam shaping [37], image optics and wave-front reconstruction [38]. In particular, the GS algorithm can also be used to compensate for turbulence distorted wavefronts in the OAM-based communication links, as proposed both theoretically and experimentally in the past years [39,40]. In our simulation, we use GS algorithm to compute a pre-compensate phase. The iterative procedure of computing the phase compensation which should load on the spatial light modulator (SLM) plane is shown in Figure 2. The specific steps of loop iteration can be described as follows:

**Step 1**. Initialization, take out the complex amplitude $a_0$ of the original LG beam distorted by turbulence and a flat phase $\varphi_0$ of Gaussian probe beam, and then perform Fast Fourier Transform (FFT) to obtain a spectral function on the image plane:

$$F\{a_0 \times \exp(i\varphi_0)\} = A_0 \times \exp(i\varphi_1), \tag{6}$$

Where $F$ represents FFT, $A_0$ and $\varphi_1$ are the amplitude and phase in the Fourier domain respectively.

**Step 2**. Replace the amplitude $A_0$ with the amplitude $B_0$ of target LG beam, but reserve the phase $\varphi_1$, the new spectral function can be obtained as:

$$U = B_0 \times \exp(i\varphi_1). \tag{7}$$

**Step 3**. Perform the Inverse Fast Fourier Transform (IFFT) and obtain the amplitude and phase function:

$$F^{-1}\{B_0 \times \exp(i\varphi_1)\} = b_0 \times \exp(i\varphi_2), \tag{8}$$

where $F^{-1}$ represents IFFT, $b_0$ and $\varphi_2$ are the amplitude and phase on the SLM plane.

**Step 4.** Continue to replace amplitude $b_0$ with the amplitude $a_0$ of the original distorted LG beam, but reserve the phase $\varphi_2$, thus the optical field $u$ on the SLM plane can be obtained as:

$$u = a_0 \times \exp(i\varphi_2). \tag{9}$$

Then, go back to step 1 to continue the loop iteration. The updated Fourier transformed phase function is substituted into the error function. According to the error precision, it can be judged that the simulation continues or jumps out of the loop iteration process. If the error condition is satisfied, jump out of the process and the phase function in this loop is the output, from which the pre-compensation phase is obtained as $\varphi_3 = -\varphi_2$.

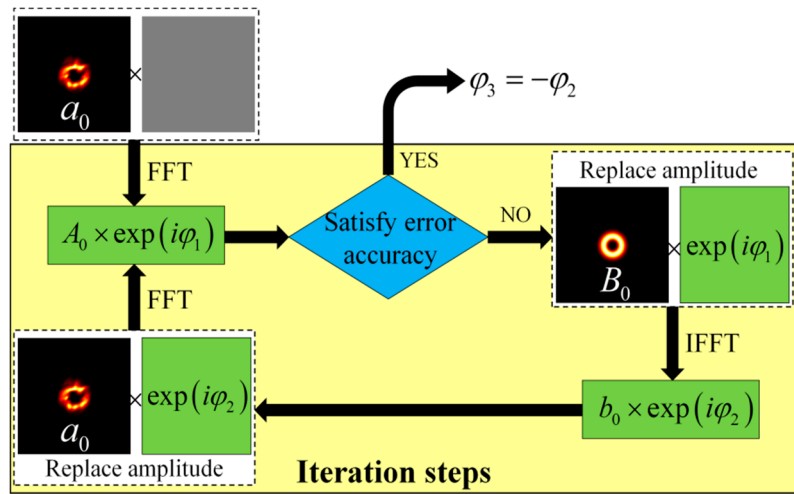

**Figure 2.** Iterative step of computing the compensation phase screen based on the Gerchberg-Saxton (GS) algorithm; (FFT: Fast Fourier Transform; IFFT: Inverse Fast Fourier Tranform).

## 3. Results

In this section, we will present the simulated retrieving performances of the OAM modes after transmitting in the different turbulences with the aids of GS algorithm, which are described by the power density (PD) defined as the ratio of the obtained target OAM mode power on the original incident OAM mode power. Based on the obtained PD spectra, we will discuss and interpret the results in details. In simulations, we set three different values of turbulence refractive-index structure parameters, including $C_n^2 = 1 \times 10^{-15} \mathrm{m}^{-2/3}$, $C_n^2 = 1 \times 10^{-14} \mathrm{m}^{-2/3}$ and $C_n^2 = 1 \times 10^{-13} \mathrm{m}^{-2/3}$, and the propagation distance $z = 1000$ m is divided into several parts with different number of random phase screen including $N = 20$, 50, and 100. The beam waist is $\omega_0 = 5$ cm with the wavelength $\lambda = 1550$ nm.

Firstly, we explore the effect of turbulent channel with a different refractive index structure parameter $C_n^2$ and phase screen number $N$ on the transmitted single OAM beam. Figure 3 shows the intensity and phase distributions of beams and the corresponding PD spectra without and with the pre-compensation of GS algorithm under simulated atmospheric turbulence with $C_n^2$ valued as $1 \times 10^{-15} \mathrm{m}^{-2/3}$ but different number of random phase screens. Sub-figures (a1, c1, e1) and (b1, d1, f1) are the concrete intensity distributions of the OAM beams, without pre-compensation, after transmitting in simulated turbulence channels with random phase screens number of $N = 20$, 50 and 100 respectively, in which the topological charge of OAM beam are $l = 3$ and $l = 5$ respectively. Sub-figures (a3, c3, e3) and (b3, d3, f3) of Figure 3 are their phases distributions respectively. Sub-figures

(a5, c5, e5) and (b5, d5, f5) of Figure 3 are their PD spectra respectively. Sub-figures (a2, c2, e2) and (b2, d2, f2), (a4, c4, e4) and (b4, d4, f4), and (a6, c6, e6) and (b6, d6, f6) of Figure 3 are the corresponding situations (intensity distributions, phases distributions, and PD spectra) with the pre-compensation of GS algorithm. The parts circled in green (a1~a6; b1~b6), blue (c1~c6; d1~d6) and red (e1~e6; f1~f6) dashed frames represent simulated turbulence channels with $N = 20$, 50 and 100 respectively. We can observe that when the refractive-index structure parameter is fixed, with increasing the number of inserted random phase screens, the damage effect of simulated turbulence on the wave-front of the transmitted OAM beam becomes more obvious.

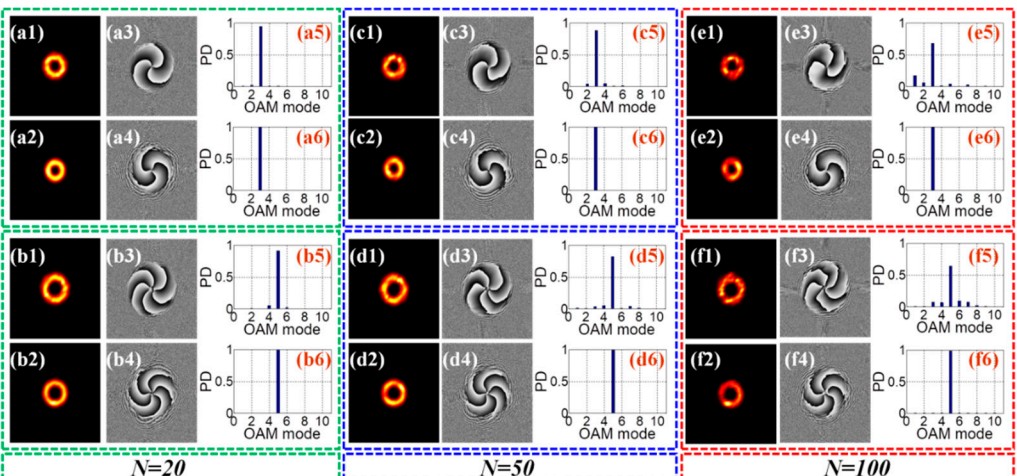

**Figure 3.** The intensity and phase distributions of orbital angular momentum (OAM) beams and their power density (PD) spectra without and with the pre-compensation of GS algorithm under simulated atmospheric turbulence with refractive index structure parameter $C_n^2$ valued as $1 \times 10^{-15} \text{m}^{-2/3}$. The parts circled in green, blue and red dashes' frames represent simulated turbulence channel with random phase screens number of $N = 20$, 50 and 100 respectively. (a1, c1, e1) and (b1, d1, f1) are intensity profiles without pre-compensation for topological charge $l = 3$ and $l = 5$ respectively, (a3, c3, e3) and (b3, d3, f3) are their corresponding phase distributions, (a5, c5, e5) and (b5, d5, f5) are their corresponding PD spectra. (a2, c2, e2) and (b2, d2, f2) are intensity profiles with the pre-compensation of GS algorithm, (a4, c4, e4) and (b4, d4, f4) are their corresponding phase distributions, (a6, c6, e6) and (b6, d6, f6) are their corresponding PD spectra.

To further analyze quantitatively, we have measured the PD spectra of the obtained LG beams with initial topological charges of $l = 3$ and $l = 5$ under turbulence with refractive index structure parameter $C_n^2 = 1 \times 10^{-15} \text{m}^{-2/3}$. For the transmitted OAM beams with topological charge of $l = 3$, without pre-compensation by GS algorithm, it shows that the PD spectra are 94.96% (Figure 3(a5)), 88.09% (Figure 4(c5)) and 67.94% (Figure 4(e5)) corresponding to different channels with $N = 20$, 50 and 100 respectively, and for $l = 5$, the corresponding PD spectra are 91.56% (Figure 3(b5)), 82.15% (Figure 3(d5)) and 64.33% (Figure 3(f5)) respectively. Then, with the aids of pre-compensation, the PDs reach 99.98% (Figure 3(a6)), 99.88% (Figure 3(c6)) and 99.46% (Figure 3(e6)) for $l = 3$, and 99.73% (Figure 3(b6)), 99.42% (Figure 3(d6)) and 98.35% (Figure 3(f6)) for $l = 5$. The concrete PD values of OAM beams without (W/o) and with (W) GS algorithm corresponding to Figure 3 have been concluded and demonstrated in Table 1. Obviously, the results demonstrate that for weak turbulence strength levels, i.e., $C_n^2 = 1 \times 10^{-15} \text{m}^{-2/3}$ (with three different phase screen numbers), the retrieval of the pre-compensation based on the GS algorithm is effective for enhancing the OAM's detecting resolution. Furthermore, the results also indicate the increasing phase screens directly affect the turbulence strength level at a fixed transmission distance. At the same time, the recovery results of the OAM beam with $l = 5$ are slightly worse than those with $l = 3$, which can be attributed to the relatively large radius of the OAM beam with higher topological charge. In general, the higher topological charge, the larger

beam's contacting area with turbulence flow; therefore, the introduced crosstalk is relatively larger, and the corresponding effect of the compensation with the GS algorithm is slightly worse.

**Table 1.** The power density (PD) values of orbital angular momentum (OAM) beams without (W/o) and with (W) GS algorithm corresponding to Figure 3.

| OAM Mode | W/o or W/ GS | N = 20 | N = 50 | N = 100 |
|---|---|---|---|---|
| $l = 3$ | W/o GS | 94.96% | 88.09% | 67.94% |
| | W/ GS | 99.98% | 99.88% | 96.46% |
| $l = 5$ | W/o GS | 91.56% | 82.15% | 64.33% |
| | W/ GS | 99.73% | 99.42% | 98.35% |

Then, as shown in Figure 4, we also investigated the influence of the increasing the turbulence level with $C_n^2$ as $1 \times 10^{-14} \text{m}^{-2/3}$. The concrete retrieving performances based on the GS algorithm have also been investigated. We can see that with increasing $C_n^2$, the transmitted intensity and phase of the OAM beams will be more divergent. Notably, when the random phase-screens number reach 100 (as shown in the third column), it is very difficult to judge the concrete topological charge according to the phase distributions alone.

The corresponding obtained PD spectrum for both $l = 3$ and $l = 5$ with and without pre-compensation under turbulence channel with $C_n^2 = 1 \times 10^{-14} \text{m}^{-2/3}$ are illustrated in Figure 4. Here, the PDs of distorted LG beam are respectively 70.26%, 62.46%, and 37.69% corresponding to phase screen number $N = 20$, 50 and 100 for $l = 3$, and 66.49%, 47.57% and 27.27% for $l = 5$. The retrieval OAM power density after the compensation of GS algorithm corresponding to $N = 20$, 50 and 100 is respectively 86.24%, 78.16% and 69.58% for $l = 3$, and 82.35%, 73.15% and 60.82% for $l = 5$. We also concluded and demonstrated the concrete PD values of OAM beams without (W/o) and with (W) GS algorithm corresponding to Figure 4, as shown in Table 2. According to these measured power spectrum distribution and concrete PD values, we find that when the refractive index structure parameter $C_n^2$ is fixed as $1 \times 10^{-14} \text{m}^{-2/3}$, with the increase of the phase screen number, the introduced crosstalk is more obvious, especially for the case of high topological charge in which the power density of the additional measured OAM mode has exceeded more than half of the power density of the target OAM mode due to crosstalk. However, after using the pre-compensation of GS algorithm, it can be seen from the recovered power spectra that the density of the additional mode can be reduced to a lower level, and we can still clearly distinguish and judge the value of the transmitted target OAM mode by setting reasonable density judgment threshold at the receiver in an OAM-based communication system. Meanwhile, by comparing Figures 3 and 4 we see that the power density spectrum under turbulence channel with phase screen number of $N = 100$ and refractive index structure parameter of $C_n^2 = 1 \times 10^{-15} \text{m}^{-2/3}$, as shown in Figure 3, is close to that with $N = 20$ and $C_n^2 = 1 \times 10^{-14} \text{m}^{-2/3}$, demonstrated in Figure 4. This means that the turbulence intensity level is directly not only related to the refractive index structure parameter, but also to the number of the simulated random phase screens; thus, under the specific combination of these two parameters, the turbulence intensity level may be similar, so the corresponding power spectrum is relatively close.

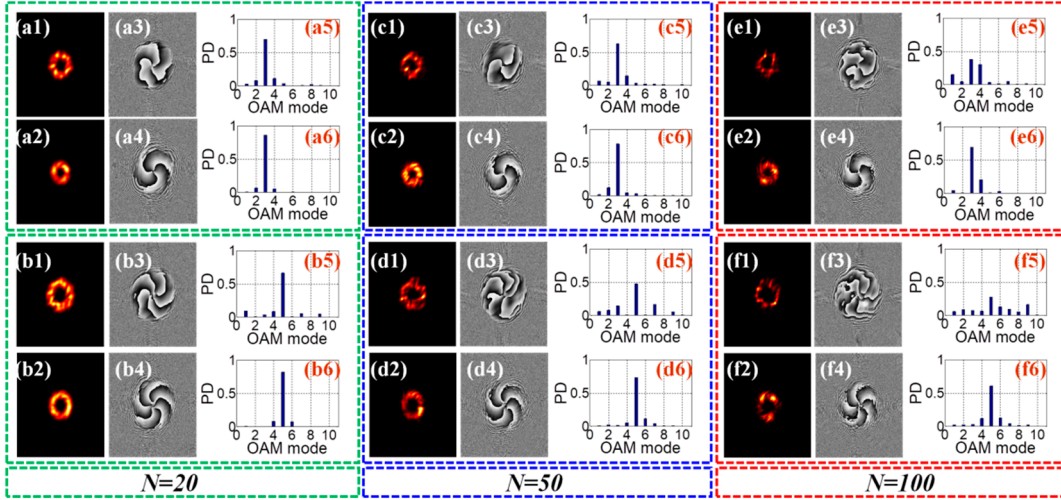

**Figure 4.** The intensity and phase distributions of OAM beams and their PD spectra without and with the pre-compensation of GS algorithm under simulated atmospheric turbulence with $C_n^2$ valued as $1 \times 10^{-14} \mathrm{m}^{-2/3}$. The parts framed in green, blue and red dashes represent simulated turbulence channel with random phase-screen number of $N = 20$, 50 and 100 respectively. (a1, c1, e1) and (b1, d1, f1) are intensity profiles without pre-compensation for OAM beams with $l = 3$ and $l = 5$ respectively, (a3, c3, e3) and (b3, d3, f3) denote their corresponding phase distributions, (a5, c5, e5) and (b5, d5, f5) show their corresponding PD spectra; (a2, c2, e2) and (b2, d2, f2) are intensity profiles with the pre-compensation of GS algorithm for OAM beams with $l = 3$ and $l = 5$ respectively, (a4, c4, e4) and (b4, d4, f4) are their corresponding phases, (a6, c6, e6) and (b6, d6, f6) their corresponding PD spectra.

**Table 2.** The PD values of OAM beams without (W/o) and with (W) GS algorithm corresponding to Figure 4.

| OAM Mode | W/o or W/ GS | N = 20 | N = 50 | N = 100 |
|---|---|---|---|---|
| $l = 3$ | W/o GS | 70.26% | 62.46% | 37.69% |
|  | W/ GS | 86.24% | 78.16% | 69.58% |
| $l = 5$ | W/o GS | 66.49% | 47.57% | 27.27% |
|  | W/ GS | 82.35% | 73.15% | 60.82% |

To further explore the impact of stronger turbulence intensity levels and find the limit that the GS algorithm can correct, we increase the value of $C_n^2$ to $1 \times 10^{-13} \mathrm{m}^{-2/3}$. As the intensity and phase distributions of transmitted LG beams illustrate in Figure 5, the beams with topological charges of $l = 3$ and $l = 5$ are all strongly damaged under the turbulence channels with three different phase screens number, not only without pre-compensation, but also with the pre-compensation of GS algorithm. And although the phase has recovered after using the GS algorithm for compensation, it seems to reach the range limit that the GS algorithm can correct.

According to the corresponding PD spectra, we find that the PDs for both distorted and retrieval with pre-compensation are all less than 50%. Obviously, although the GS algorithm is used to compensate the light intensity and phase, when the value of $C_n^2$ is large, the turbulence intensity will reach a strong level, and it is beyond the compensation limit range of the GS algorithm, which can also be inferred from Table 3. Therefore, according to the distribution of the power density spectrum, we are unable to accurately detect the OAM mode of the actual transmitted target LG beam.

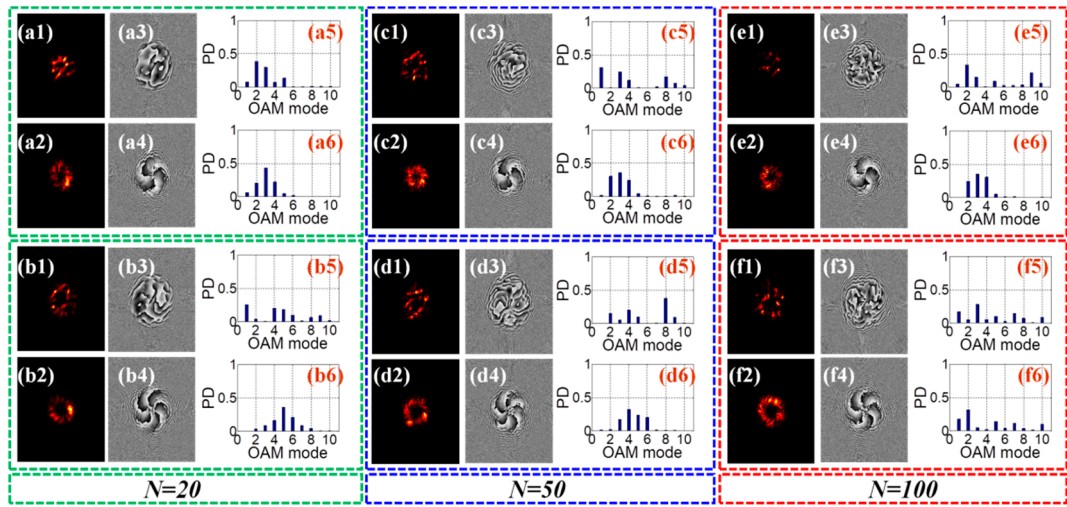

**Figure 5.** The intensity and phase distributions of OAM beams and their PD spectra without and with the pre-compensation of GS algorithm under simulated atmospheric turbulence with refractive-index structure parameter $C_n^2$ valued as $1 \times 10^{-13} \mathrm{m}^{-2/3}$. The parts framed in green, blue and red dashes represent random phase screens number $N = 20$, 50 and 100 respectively. (a1, c1, e1) and (b1, d1, f1) are intensity profiles without pre-compensation for topological charge $l = 3$ and $l = 5$ respectively, (a3, c3, e3) and (b3, d3, f3) are their corresponding phase, (a5, c5, e5) and (b5, d5, f5) are their corresponding PD spectra; (a2, c2, e2) and (b2, d2, f2) are intensity profiles with the pre-compensation of GS algorithm for $l = 3$ and $l = 5$ respectively, (a4, c4, e4) and (b4, d4, f4) demonstrate their corresponding phase distributions, (a6, c6, e6) and (b6, d6, f6) are their corresponding PD spectra accordingly.

We have explored the case of transmitting LG beam with single OAM state. In the actual OAM-based optical communication system, based on the orthogonal property, the multiplexing vortex beam with multiple OAM modes (topological charges) is usually adopted to represent more data bits. Here, we have also explored the case of transmitting coherent multiplexing LG beams under different turbulence channel by changing the refractive index structure parameter $C_n^2$ from $1 \times 10^{-15} \mathrm{m}^{-2/3}$ to $1 \times 10^{-13} \mathrm{m}^{-2/3}$ and the simulated random phase screen number $N$ from 20 to 100. In our simulations, two LG beams with topological charges of $l = 2$ and $l = 6$ are adopted for coherent multiplexing. The results under turbulence with refractive index structure parameter $C_n^2 = 1 \times 10^{-15} \mathrm{m}^{-2/3}$ are shown in Figure 6.

**Table 3.** The PD values of OAM beams without (W/o) and with (W) GS algorithm corresponding to Figure 5.

| OAM Mode | W/o or W/ GS | N = 20 | N = 50 | N = 100 |
|---|---|---|---|---|
| $l = 3$ | W/o GS | 27.73% | 24.47% | 15.16% |
| | W/ GS | 43.43% | 35.89% | 35.08% |
| $l = 5$ | W/o GS | 18.76% | 10.60% | 10.22% |
| | W/ GS | 36.17% | 23.31% | 13.84% |

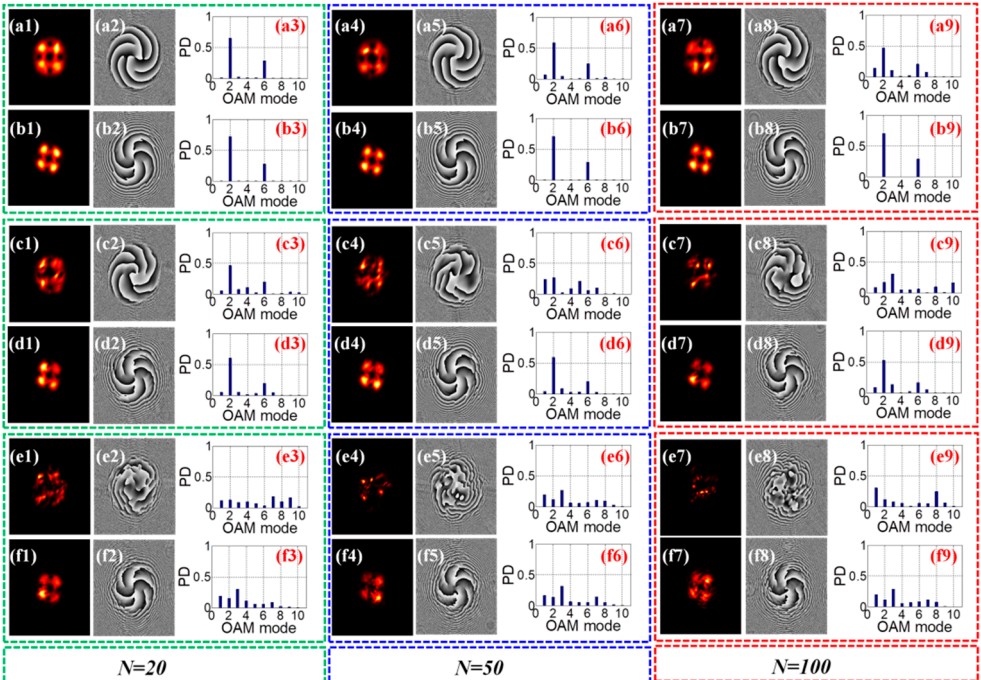

**Figure 6.** The intensity and phase distributions of multiplexing OAM beams (*l* = 2 and *l* = 6) and their PD spectra without and with the pre-compensation of GS algorithm under simulated atmospheric turbulence with different refractive-index structure parameter of $C_n^2$ valued as $1 \times 10^{-15} \text{m}^{-2/3}$, $1 \times 10^{-14} \text{m}^{-2/3}$ and $1 \times 10^{-13} \text{m}^{-2/3}$ (corresponding to the first, second and third horizontal line). The parts framed in green, blue and red dashes represent random phase screens number *N* = 20, 50 and 100 respectively. (a1, a4, a7), (c1, c4, c7) and (e1, e4, e7) are intensity profiles without pre-compensation, (a2, a5, a8), (c2, c5, c8) and (e2, e5, e8) demonstrate their corresponding phase distributions, (a3, a6, a9), (c3, c6, c9) and (e3, e6, e9) show their corresponding PD spectra; (b1, b4, b7), (d1, d4, d7) and (f1, f4, f7) are intensity profiles with the pre-compensation of GS algorithm; (b2, b5, b8), (d2, d5, d8) and (f2, f5, f8) depict their corresponding phase distributions, (b3, b6, b9), (d3, d6, d9) and (f3, f6, f9) correspond to their PD spectra.

Sub-figures (a1, a4, a7), (c1, c4, c7) and (e1, e4, e7) of Figure 6 are their intensity distributions under the turbulence channel with three different simulated random phase screens without pre-compensation. Sub-figures (a2, a5, a8), (c2, c5, c8) and (e2, e5, e8) of Figure 6 are their corresponding phase distributions. Sub-figures (a3, a6, a9), (c3, c6, c9) and (e3, e6, e9) of Figure 6 are their corresponding PD spectra. Sub-figures (b1, b4, b7), (d1, d4, d7) and (f1, f4, f7), Sub-figures (b2, b5, b8), (d2, d5, d8) and (f2, f5, f8) and Sub-figures (b3, b6, b9), (d3, d6, d9) and (f3, f6, f9) of Figure 6 are the intensity and phase distributions and the PD spectra results with the pre-compensation of GS algorithm. And the PD values of OAM beams without (W/o) and with (W) GS algorithm corresponding to Figure 6 have demonstrated in Table 4. It can be seen that the distortion of multiplexing LG beam increases gradually when the refractive index structure parameter of $C_n^2$ increases with the fixed phase screen number, which is similar to that of the single OAM beam. Meanwhile, with increasing phase screen number, the pre-compensation result is similar when the refractive structure parameter of $C_n^2$ is fixed. From the PD spectra and the concrete values shown in Figure 6 and Table 4 respectively, when the value of $C_n^2$ is no more than $1 \times 10^{-14} \text{m}^{-2/3}$, no matter how many the simulated random phase screens inserted, the pre-compensation of the GS algorithm can effectively reduce the crosstalk, and the OAM mode value of the transmitted multiplexing LG beam can be effectively judged according to the concrete spectrum distribution. However, when the value of $C_n^2$ increase to $1 \times 10^{-13} \text{m}^{-2/3}$, which reaches a strong turbulence strength level, the introduced crosstalk is relatively larger and exceeds the compensation limit range of the GS algorithm; thus, the OAM mode cannot be accurately detected.

**Table 4.** The PD values of OAM beams without (W/o) and with (W) GS algorithm corresponding to Figure 6.

| $C_n^2$ Value | OAM Mode | W/o or W/ GS | N = 20 | N = 50 | N = 100 |
|---|---|---|---|---|---|
| $1 \times 10^{-15} \mathrm{m}^{-2/3}$ | $l = 2$ | W/o GS | 65.10% | 58.70% | 46.88% |
| | $l = 6$ | | 28.43% | 25.03% | 20.80% |
| | $l = 2$ | W/ GS | 72.29% | 70.65% | 70.60% |
| | $l = 6$ | | 27.67% | 29.03% | 29.01% |
| $1 \times 10^{-14} \mathrm{m}^{-2/3}$ | $l = 2$ | W/o GS | 46.58% | 26.60% | 17.02% |
| | $l = 6$ | | 19.82% | 6.08% | 6.13% |
| | $l = 2$ | W/ GS | 60.69% | 59.07% | 52.63% |
| | $l = 6$ | | 19.63% | 20.03% | 16.58% |
| $1 \times 10^{-13} \mathrm{m}^{-2/3}$ | $l = 2$ | W/o GS | 12.57% | 11.84% | 11.97% |
| | $l = 6$ | | 3.04% | 7.10% | 5.57% |
| | $l = 2$ | W/ GS | 15.56% | 13.95% | 11.32% |
| | $l = 6$ | | 5.93% | 5.27% | 8.49% |

In addition, by comparing the above case of transmitting single OAM beam with that of multiplexing LG beam, we can find that, for the transmission of single OAM beam, under the same turbulence strength level, the PD of the target OAM mode after retrieving by the GS algorithm is significantly higher than that of the crosstalk channel. And when the turbulence strength does not exceed the level corresponding to the refractive index structure parameter of $C_n^2 = 1 \times 10^{-13} \mathrm{m}^{-2/3}$ with the number of simulated random phase screens $N = 20$, the GS algorithm can effectively compensate the distorted LG beam and the transmitted target OAM mode can be precisely determined according to the PD spectra distributions. However, for the transmission of multiplexing LG beam, since multiple channels will introduce more internecine crosstalk, especially when the interval between different OAM modes is not very large [18], when the $C_n^2$ only increases to $1 \times 10^{-14} \mathrm{m}^{-2/3}$ with $N = 100$, the limit range that can be corrected by the GS algorithm is reached. This indicates that the retrieving performance of transmitting single LG beam is better than that of transmitting multiplexing LG beam. Simultaneously, we can conclude that for the transmission system in simulation, the maximum turbulence strength level that can be corrected by the GS algorithm is the level corresponding to the refractive index structure parameter valued as $C_n^2 = 1 \times 10^{-14} \mathrm{m}^{-2/3}$ with the random phase screen number valued as $N = 100$.

To further investigate the limitation of the phase screen number, we have simulated the case of turbulence channel with phase screen number of $N = 150$. The corresponding performances of LG beams ($l = 5$) with and without pre-compensation of GS algorithm under simulated atmospheric turbulence with $C_n^2 = 1 \times 10^{-15} \mathrm{m}^{-2/3}$ and $C_n^2 = 1 \times 10^{-14} \mathrm{m}^{-2/3}$ are shown in Figure 7. This shows that the PD spectra of distorted LG beams are 54.80% and 14.66% respectively for $C_n^2 = 1 \times 10^{-15} \mathrm{m}^{-2/3}$ and $C_n^2 = 1 \times 10^{-14} \mathrm{m}^{-2/3}$. And the retrieval OAM PDs after the compensation of GS algorithm corresponding to $C_n^2 = 1 \times 10^{-15} \mathrm{m}^{-2/3}$ and $C_n^2 = 1 \times 10^{-14} \mathrm{m}^{-2/3}$ can reach 97.51% and 19.32% respectively. It Obviously indicates that, under the weak turbulence strength level of $C_n^2 = 1 \times 10^{-15} \mathrm{m}^{-2/3}$, the retrieval of the pre-compensation based on the GS algorithm is effective for enhancing the OAM's detecting resolution. In contrast, for the case of turbulence strength level of $C_n^2 = 1 \times 10^{-14} \mathrm{m}^{-2/3}$ with phase screen number of $N = 150$, it obviously closes to strong turbulence strength level; thus, the GS algorithm cannot effectively retrieve the distorted LG beams, which is also the same as the obtained conclusion earlier.

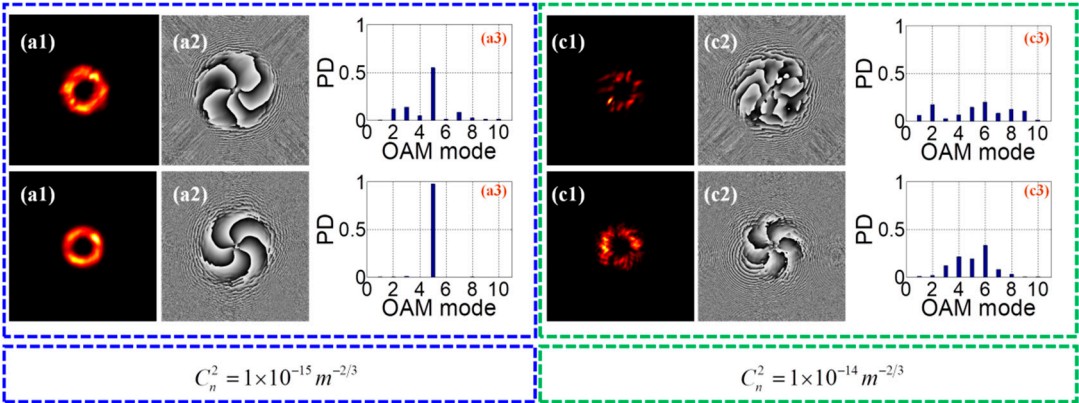

**Figure 7.** Intensity, phase and power density spectrum distribution of OAM beams ($l = 5$) without and with the pre-compensation of GS algorithm under simulated atmospheric turbulence with different refractive index structure parameter $C_n^2$ valued as $1 \times 10^{-15}$m$^{-2/3}$, $1 \times 10^{-14}$m$^{-2/3}$ for phase screens number $N = 150$. (a1, a2, a3) and (c1, c2, c3) are respectively intensity, phase and power density spectrum without pre-compensation, (b1, b2, b3) and (d1, d2, d3) are their corresponding are power density with pre-compensation.

## 4. Conclusions

In summary, we have described fundamental concept and principle of the LG beams carrying OAM, the turbulence model and the GS algorithm, and emulated the atmospheric turbulence by inserting random phase screens on the transmission path. We have also demonstrated the distortion effect of different turbulence strength levels by changing refractive-index structure parameter of $C_n^2$ and the number of phase screens of $N$ on the transmitting LG beams. Then, based on GS algorithm, we calculate a pre-compensation phase to compensate the distorted LG beams, and give the corresponding comparisons and discussions of performances with and without pre-compensation for transmitting single and multiplexing OAM beam. The results illustrate that the parameters $C_n^2$ and $N$ directly determine the level of the turbulence strength, and when the turbulence strength level is selected within the correction limit range of $C_n^2 = 1 \times 10^{-14}$m$^{-2/3}$ with $N = 100$, the LG beam can be effectively retrieved and the OAM can be directly detected by adopting GS algorithm and observing the PD spectra. In addition, according to a comparison of performance results, the retrieving performance of transmitting single OAM beam is better than that of transmitting multiplexing OAM beam.

**Author Contributions:** Conceptualization, M.I.D. and Z.W.; methodology, K.G.; software, F.S. and H.Z.; formal analysis, M.I.D., Z.W. and Y.S.; writing—original draft preparation, M.I.D. and Z.W.; writing—review and editing, Z.G., J.G., R.S. and Z.D.; supervision, Z.G.; funding acquisition, Z.G.

**Funding:** National Natural Science Foundation of China (61775050, 11804073), Fundamental Research Funds for the Central Universities (JZ2018HGBZ0309, JZ2018HGTB0240).

**Conflicts of Interest:** The authors declare no conflict of interest.

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
