# Peer review of "Retrieving Performances of Vortex Beams with GS Algorithm after Transmitting in Different Types of Turbulences"

_applsci, doi:10.3390/app9112269_

Round 1
Reviewer 1 Report
This paper indicates good results behavior of receive OAM beams.
I require some minor revises as following.
GS algorithm should be appeared in title.
Figures 3~6 are too small to read. Improvement required.
This paper only shows results of GS algorithm and limit range of this algorithm. However, authors introduce other compensation methods, MIMO eualization, adaptive optics, and so on. Are there any possibility to enhance limit range by using other methods?
If authors have any ideas or comments, please add explanation about this problem.
Author Response
See the uploaded file.

Reviewer 2 Report
This paper theoretically investigates the effect of atmospheric turbulence on the transmission of single and multiplexed vortex beams with orbital angular momenta (OAM) and explores the performance of the Gerchberg-Saxton algorithm on the retrieval of OAM modes’ information from distorted beams’ amplitude.
The remarkable points of this paper are as follows:
• A high level of numerical analysis is performed and the results of simulations are encompassing various combinations of parameters in the Kolmogorov turbulence model.
• The figures summarizing the results are presented in a comprehensive manner with a nice combination of information of transmitted beams.
• The presence of a parameter "turbulence intensity" implicitly- suggested is pointed out, and the authors attempt to clarify its characteristics at a semi-quantitative level as a function of the refractive-index structure parameter and the number of random phase screen in Kolmogorov's model of turbulence.
The subject on the retrieval of OAM modes' information from distorted vortex beams after transmission in turbulences is practically very important and would attract great interest of a broad readership. This work is suitable for Applied Sciences, while there remain several improvable points listed below.
• Although the present combination of the sub-figures in each figure is very attractive, the numbering (or naming) of them is easy to get confused and not so kind to readers. Any improvement to the numbering/naming would be appreciated.
• The range of simulated situations and the discussion on the results are extremely rich. I think the readability of the manuscript would be greatly improved if the important points of the results are summarized in tables.
I recommend the publication of this paper after authors’ reconsideration on the points listed above and minor revision of the manuscript.
Author Response
See the uploaded file.
